# HGSOLVER: POSITION-ENHANCED PHYSICS ATTENTION INFORMED HETEROGENEOUS GEOMETRIES NEURAL SOLVER FOR PDES

## ABSTRACT

Partial differential equations (PDEs) provide a fundamental framework for modeling complex physical phenomena. However, modeling PDEs on heterogeneous geometries remains a significant challenge for both traditional numerical solvers and neural operator methods, as sparse observations, multiphysics interactions, and distinct discretizations often produce heterogeneous geometries between the observation and output spaces. In this work, we introduce a unified perspective on physics attention, formulating physical states as projections of observation embeddings onto learnable functional bases in Hilbert space. Building on this formulation, we introduce a position-enhanced physics attention mechanism that incorporates coordinate representations of these bases via rotary position embeddings, thereby enabling more effective modeling of heterogeneous interactions. Leveraging this mechanism, we develop HGsolver, an encoder–decoder framework designed for PDE tasks on heterogeneous domains. Extensive experiments demonstrate that HGsolver achieves state-of-the-art performance across forward, inverse, and reconstruction benchmarks under heterogeneous geometries, while a minimally modified variant, TransolverXP, also delivers competitive results on standard homogeneous benchmarks. These findings highlight the importance of effective interactions among physical states in advancing neural PDE solvers and their potential to address the complexity of the heterogeneous real-world geometries.

## 1 INTRODUCTION

As a torch, partial differential equations (PDEs) establish a foundational framework for humanity to gain insight and conquer the physical world through the lens of mathematical models within the realm of scientific inquiry. Analytically solving most PDEs is infeasible, and conventional numerical solvers(Zhang et al., 2021) are usually confined to specific classes of PDEs, necessitating customization for each new formulation. This task-specific nature often leads to limited generality, reduced flexibility, and substantial computational overhead, particularly when handling complex geometries(Umetani & Bickel, 2018) or performing repeated simulations under varying conditions(Umetani & Bickel, 2018). In recent years, deep neural network surrogates, particularly neural operators (Li et al., 2020; 2023c), have emerged as a computationally efficient paradigm for advancing the modeling and approximation of complex PDEs. Leveraging their strong nonlinear modeling capability(Lu et al., 2021), they can learn input–output mappings of PDE-governed tasks from data and infer solutions much faster than numerical methods.

PDEs are typically discretized into large-scale meshes with complex geometries to enable precise simulations. Several backbones(Hao et al., 2023; Li et al., 2023b) leveraging individual point features have been proposed to address these challenges. However, these point-based architectures encounter significant difficulties in capturing the intricate physical correlations in PDEs, particularly in industrial design involving extremely large and complex geometries, which typically feature highly coupled multiphysics interactions(Trockman & Kolter, 2022).

To this end, Transolver(Wu et al., 2024) introduces a state-based attention mechanism that can operate under arbitrary general geometries. Nevertheless, it fails to capture the heterogeneity of interactions among state tokens, especially those derived from distinct discretizations. In real-world scenar-

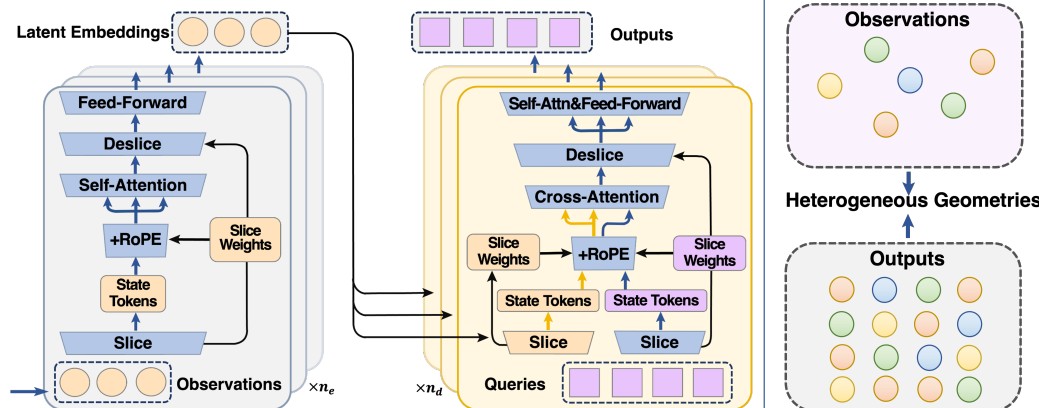

Figure 1: Overview of the **HGsolver** architecture (**left**) and representative illustrations of heterogeneous geometries (**right**).

ios, the available input is often sparse in both spatial and temporal domains, since experimental data are typically collected using a limited set of sensors or measurement devices. Consequently, inputs and outputs frequently come from heterogeneous geometries, while sufficiently complete outputs are crucial for providing accurate guidance in complex applications such as computational fluid dynamics for car and airplane design. Several diffusion-based methods(Gao & Ji, 2019; Huang et al., 2024), which exploit robust generative capabilities, have also been proposed to address this challenge. However, these approaches suffer from substantial drawbacks, including prolonged training time, slow inference, and an excessive number of parameters. These limitations highlight the urgent need for methods that can more effectively handle problems defined on heterogeneous geometries.

In this paper, we present a unified and novel perspective on physics attention, formulated in a setting where physical states are represented as projections of input embeddings onto a set of well-learned functional bases in Hilbert space. Consequently, the distinctions between different physical states are determined by the variations among these learned bases. Building on this foundation, the learned functional bases can be represented by coordinates in Hilbert space, which provide a more precise characterization of their relative positions, particularly when the functional bases are discretized over heterogeneous geometries. To this end, we propose a position-enhanced physics attention mechanism that integrates these coordinate representations into all physical state tokens through rotary position embeddings (Su et al., 2021). As illustrated in Fig. 1, we further introduce **HGsolver**, a framework specifically designed to address PDE tasks on heterogeneous geometries such as super-pixel and restructured domains. With the proposed position-enhanced physics attention, **HGsolver** efficiently captures complex heterogeneous interactions.

To sum up, our main contributions are as follows:

(i) From the perspective of projecting embeddings onto learnable bases in Hilbert space, we offer a novel insight into physical states and the mechanism of physics attention.

(ii) Building upon the positional representations of the learnable bases, we introduce a position-enhanced physics attention mechanism that more effectively captures interactions among physical states, particularly for those discretized over heterogeneous geometries.

(iii) Leveraging cross position-enhanced physics attention, we design **HGsolver** based on an encoder-decoder architecture, enabling it to tackle tasks involving heterogeneous geometries between inputs and outputs, such as sparse prediction and super-pixel domains.

(iv) **HGsolver** achieves sota performance on benchmarks involving heterogeneous geometries, including forward, inverse, and reconstruction problems of PDEs, while a minimally modified variant **TransolverXP**, also remains competitive on standard benchmarks.

## 2 METHODOLOGY

In this section, we present a novel perspective on physics attention and an effective approach for its enhancement. We begin by reviewing the conventional implementations of physics attention, and then provide a comprehensive description of our methodology and the improvements it introduces.

## 2.1 PROBLEM SETUP

Our study focuses on two fundamental categories of PDEs: static PDEs and time-dependent (dynamic) PDEs. Static PDEs, such as Darcy's law (Hubbert, 1956) and models that describe the behavior of solid materials(Dym et al., 1973), are employed to characterize the equilibrium states of physical systems. In contrast, time-dependent PDEs model the temporal evolution of physical systems, such as the Navier–Stokes equations governing fluid dynamics(McLean, 2012).

In this paper, we investigate PDEs defined on a domain $\Omega \subset \mathbb{R}^d \times [0, T]$ , with the solution function space $\mathcal{H}$ over $\Omega$. The objective is to learn an operator $\mathcal{G}$ that maps the input function space $\mathcal{A}$ to the solution space $\mathcal{H}$, i.e., $\mathcal{G} : \mathcal{A} \to \mathcal{H}$ and $\mathcal{G}(a) = u \in \mathcal{H}$ is the solution function over $\Omega$.

To learn the operator $\mathcal{G}$, neural operator model is trained using a dataset $\mathcal{D} = \{(a_k, u_k)\}_{1 \le k \le D}$, where $u_k = \mathcal{G}(a_k)$. In practice, $a_k$ is often discretized on the mesh $\{\boldsymbol{x}_i \in \Omega\}_{1 \le i \le N}$. The discretized representation $a_k$ is given by $\{(\boldsymbol{x}_i, \boldsymbol{a}_k^i)\}_{1 \le i \le N}$, where $\boldsymbol{a}_k^i = a_k(\boldsymbol{x}_i)$. In this manner, the input functions $a_k$ are represented by $\{(\boldsymbol{x}_i, \boldsymbol{a}_k^i)\}_{1 \le i \le N}$.

For the corresponding solution function $u_k$, we discretize it on the mesh $\{\boldsymbol{y}_i \in \Omega\}_{1 \le i \le N'}$, with the discretized representation $u_k$ given by $\{(\boldsymbol{y}_i, \boldsymbol{u}_k^i)\}_{1 \le i \le N'}$, where $\boldsymbol{u}_k^i = u_k(\boldsymbol{y}_i)$. To model the operator, $\mathcal{G}$ is parameterized by a neural network $\tilde{\mathcal{G}}_w$, which takes the pair $(\boldsymbol{x}_i, \boldsymbol{a}_k^i)$ as input, and outputs $\tilde{\mathcal{G}}_w \left( (\boldsymbol{x}_i, \boldsymbol{a}_k^i), \boldsymbol{y}_i \right) = \{\tilde{\boldsymbol{u}}_k^i\}_{1 \le i \le N'}$ on mesh $\{\boldsymbol{y}_i\}_{1 \le i \le N'}$ , which serves as an approximation to $u_k$. The goal is to minimize the mean squared error (MSE) loss between the predicted solution and the true data, as given by

$$\min_{w \in W} \frac{1}{D} \sum_{k=1}^{D} \frac{1}{N'} \left\| \tilde{\mathcal{G}}_w \left( (\boldsymbol{x}_i, \boldsymbol{a}_k^i), \boldsymbol{y}_i \right) - \{u_k^i\}_{1 \le i \le N'} \right\|_2^2 \tag{1}$$

where $w$ denotes the set of network parameters and $W$ represents the space of $w$. Detailed structure of $\tilde{\mathcal{G}}_w$ will be introduced in following sections.

## 2.2 IMPLEMENTATION OF PHYSICS ATTENTION

While attention mechanisms(Vaswani et al., 2017) have demonstrated remarkable effectiveness in both computer vision (CV)(Khan et al., 2022) and natural language processing (NLP)(Brown et al., 2020), their application to neural PDE solvers remains challenging. In particular, attention-based approaches often struggle to faithfully capture the complex physical correlations within the computational domain $\Omega$, which is typically discretized into a set of mesh points $\{\boldsymbol{z}_i\}_{i=1}^{N}$. These mesh points, being finite samples of the underlying continuous physical space, motivate the need to learn the intrinsic physical states. However, the large number of discretized points in the mesh can overwhelm the attention mechanism, making it challenging to identify reliable correlations. To address this, physics-based attention(Wu et al., 2024), by focusing on physics-sensitive regions, proves to be particularly effective in capturing the underlying physical information within complex and high-dimensional meshes.

Given a mesh with $N$ nodes on positions $\boldsymbol{X} = \{\boldsymbol{x}_i\}_{i=1}^{N}$, to capture the intrinsic physical interactions under the challenging mesh configurations, Physics-Attention first assigns $C$-channel input embeddings $\boldsymbol{Z} = \{\boldsymbol{z}_i\}_{i=1}^{N} \in \mathbb{R}^{N \times C}$ to $M$ physical states $\boldsymbol{S} = \{\boldsymbol{s}_j\}_{j=1}^{M} \in \mathbb{R}^{M \times C}$, based on the learned slice weights $\boldsymbol{W} = \{\boldsymbol{w}_i\}_{i=1}^{N} \in \mathbb{R}^{N \times M}$ from the inputs. Each weight vector $\boldsymbol{w}_i \in \mathbb{R}^{1 \times M}$ represents the degree that point $\boldsymbol{x}_i$ with embeddings $\boldsymbol{z}_i$ belongs to each physical state. Specifically, the physical states are aggregated from all point representations based on the learned slice weights, which can be formalized as:

$$\text{Physical States: } \boldsymbol{S} = \{\boldsymbol{s}_j\}_{j=1}^{M} = \left\{ \frac{W_{\cdot,j}^{\top} \boldsymbol{Z}}{W_{\cdot,j}^{\top} \mathbf{1_N}} \right\}_{j=1}^{M}, \tag{2}$$

The success of Physics-Attention hinges on the assumption of homogeneous physical states. Transolver utilizes the $\mathrm{Softmax}$ function to compute the slice weights, thereby ensuring a sharper distribution. However, the phenomenon of degeneration persists as the model depth increases. To mitigate this issue, Transolver++(Luo et al., 2025) introduces a local adaptive mechanism that employs the Gumbel-Softmax(Jang et al., 2016) for differentiable sampling from the discrete categorical distribution, and sets the temperature parameter $\tau_0$ in the Gumbel-Softmax function learnable, as outlined

in Eq. (3).

$$\text{Rep-Slice}: \boldsymbol{W} = \{\boldsymbol{w}_i\}_{i=1}^N = \left\{\text{Softmax}\left(\frac{\text{Linear}(\boldsymbol{z}_i) - \log(-\log \epsilon_i)}{\tau_i}\right)\right\}_{i=1}^N \tag{3}$$

$$\text{Ada-Temp}: \boldsymbol{\tau} = \{\tau_i\}_{i=1}^N = \{\tau_0 + \text{Linear}(\boldsymbol{z}_i)\}_{i=1}^N,$$

where $\tau_0$ is the temperature constant and $\boldsymbol{\epsilon} = \{\epsilon_i\}_{i=1}^N, \epsilon_i \sim \mathcal{U}(0,1)$. Here $\log(-\log \epsilon_i) \sim$ Gumbel$(0,1)$, where Gumbel is a type of generalized extreme value distribution. After that, a vanilla attention mechanism is applied on $\boldsymbol{S} = \{\boldsymbol{s}_j\}_{j=1}^M$ to capture intricate correlations among different states $\{\boldsymbol{s}_j\}_{j=1}^M$ as follows.

$$\boldsymbol{Q}, \boldsymbol{K}, \boldsymbol{V} = \text{Linear}(\boldsymbol{S}), \ \ \boldsymbol{S}' = \text{Softmax}\left(\frac{\boldsymbol{Q}\boldsymbol{K}^\top}{\sqrt{C}}\right)\boldsymbol{V}. \tag{4}$$

Finally, Physics-Attention applies the deslice operation to project the updated states $\{\boldsymbol{s}_j'\}_{j=1}^M$ back onto the mesh representation $\{\boldsymbol{z}_i'\}_{i=1}^N$ through slice weights as Eq. (5). This operation establishes a learnable non-linear mapping between $\boldsymbol{Z}$ and $\boldsymbol{Z}'$. Consequently, Physics-Attention can be flexibly stacked into more complex architectures like transformer, thereby enhancing its ability to capture the underlying physical correlations.

$$\boldsymbol{Z}' = \{\boldsymbol{z}_i'\}_{i=1}^N = \{\sum_{j=0}^M \boldsymbol{w}_{i,j}\boldsymbol{s}_j'\}_{i=1}^N \tag{5}$$

### 2.3 A Projection Perspective of Physics Attention

As shown in Eq. (2), the $j$-th physical state can be expressed as $\boldsymbol{s}_j = \boldsymbol{v}_j\boldsymbol{Z}/N$, where $\boldsymbol{v}$ is defined as $\boldsymbol{v}_j := (N W_{\cdot,j}^\top)/(W_{\cdot,j}^\top \mathbf{1_N}) \in \mathbb{R}^{1 \times N}$. Here, $\boldsymbol{v}_j := \boldsymbol{v}(\boldsymbol{Z}, \theta_j)$ is parameterized by the learnable parameters $\theta_j$ and the input embeddings $Z \in \mathbb{R}^{N \times C}$ as described in Eq. (3). The input embedding $Z_{i,\cdot}$ encodes the latent features at position $\boldsymbol{x}_i$. The normalized slice weights $v_{j,i}$ quantify the fraction of features that the state $\boldsymbol{s}_j$ receives from position $\boldsymbol{x}_i$. Consequently, for a given position $\boldsymbol{x}_k$, one can identify the corresponding input embedding $Z_k$ and the slice weights $v_{j,k}$ defined at position $\boldsymbol{x}_i$. Thus, the quantities $\boldsymbol{Z}$ and $\boldsymbol{v}(\boldsymbol{Z}, \theta_j)$ can be equivalently represented in a functional form as shown in Eq. (6), with the corresponding state $\boldsymbol{s}_j$ expressed as in Eq. (7).

$$\kappa_{\{\boldsymbol{Z}, \theta_j\}} : \Omega \to \mathbb{R}^1, \text{where} \ \ \kappa_{\{\boldsymbol{Z}, \theta_j\}}(\boldsymbol{x}_i) = v_{j,i} \ \ \ \ i = 1, 2, \ldots N$$
$$h : \Omega \to \mathbb{R}^C, \text{where} \ \ h(\boldsymbol{x}_i) = Z_{\{i,\cdot\}} \ \ \ \ \ \ \ \ \ \ \ i = 1, 2, \ldots N \tag{6}$$

$$\boldsymbol{s}_j = \frac{1}{N}\boldsymbol{v}\boldsymbol{Z} = \frac{1}{N}\sum_{i=1}^N \kappa_{\{\boldsymbol{Z}, \theta_j\}}(\boldsymbol{x}_i)h(\boldsymbol{x}_i), \ \ \ \ j = 1, 2, \ldots M \tag{7}$$

In the following sections, we simplify the notation by rewriting $\kappa_{\boldsymbol{Z}, \theta_j}$ as $\kappa_j$ for clarity. As demonstrated in Eq. (8), the physical state can be interpreted as a numerical approximation of an integral of the product of slice weights and input embeddings.

$$\boldsymbol{s}_j = \frac{1}{N}\sum_{i=1}^N \kappa_j(\boldsymbol{x}_i)h(\boldsymbol{x}_i) \approx \frac{1}{\|\Omega\|}\int_\Omega \kappa_j(\xi)h(\xi)\mathrm{d}\xi = \frac{\langle \kappa_j, h\rangle}{\|\Omega\|}, \ \ j = 1, 2, \ldots M \tag{8}$$

It worth to note that the physical state $\boldsymbol{s}_j$ can be interpreted as the inner product between the input function $h$ and a learnable functional basis $\kappa_j$ in a Hilbert space. Without loss of generality, we set $\|\Omega\| = 1$ for simplicity. Moreover, if we apply an $L^2$ regularization on $\boldsymbol{W}$, defining as:

$$v_j = \{\kappa_j(x_i)\}_{i=1}^N = (N W_{\cdot,j}^\top)/(\sqrt{W_{\cdot,j}^\top W_{\cdot,j}}), \ \ j = 1, 2, \ldots M \tag{9}$$

Then, $\kappa_j$ becomes a unit vector satisfying $\langle \kappa_j, \kappa_j\rangle = 1$ in the Hilbert space $L^2(\Omega)$. Consequently, the physical state $\boldsymbol{s}_j$ can be interpreted as the projection of the input function $h$ onto the learnable functional basis $\kappa_j$ where $\boldsymbol{s}_j = \langle \kappa_j, h\rangle/\langle \kappa_j, \kappa_j\rangle$.

Moreover, the unnormalized physics attention score between the $m$-th and $n$-th states, prior to applying the $\mathrm{Softmax}$ operation, can be expressed as Eq. (10). Here, ATT denotes the vanilla attention functional score in the original space $h$ discretized as $N$ tokens. The attention score is computed by first projecting the state representations onto query and key vectors, where $\boldsymbol{W}_q$ and $\boldsymbol{W}_k$ are learnable weight matrices associated with the query and key transformations, respectively.

$$
\begin{aligned}
\text{P-ATT}_{m,n} &= C^{-1/2}\langle \kappa_m, h\boldsymbol{W}_q\rangle^\top \langle \kappa_n, h\boldsymbol{W}_k\rangle \\
&= C^{-1/2}\iint \kappa_m(\xi_1)h(\xi_1)\boldsymbol{W}_q\boldsymbol{W}_k^\top h(\xi_2)^\top \kappa_n(\xi_2)\mathrm{d}\xi_1\mathrm{d}\xi_2 \\
&= C^{-1/2}\iint \kappa_m(\xi_1)\text{ATT}(\xi_1,\xi_2)\kappa_n(\xi_2)\mathrm{d}\xi_1\mathrm{d}\xi_2 = C^{-1/2}\langle \kappa_m, \kappa_n\rangle_{\text{ATT}}
\end{aligned}
\tag{10}
$$

As shown in Eq. (10), the attention score in the physics attention mechanism can be interpreted as the inner product of learnable basis functions $\kappa$, weighted by the vanilla attention score, which is computed over the original space with $N$ tokens. To summarize, we now present a projection-based perspective of physical states and a weighted inner-products interpretation on the operation of physics attention. In this interpretation, the $\mathrm{slice}$ component is designed to learn a set of essential bases from the input functional $h$. By projecting $h$ onto these learned bases, we obtain distinct physical states. Computing the inner products of the learned bases, modulated by canonical attention scores, effectively captures the interactions among different physical states. However, vanilla Physics Attention primarily attends to the scalar magnitudes of these projections while disregarding the geometric orientations of the bases. As a consequence, distinct bases $\kappa_1$ and $\kappa_2$, corresponding to different physical subspaces of $h$, become indistinguishable from the perspective of other states whenever $\langle \kappa_1, h\rangle = \langle \kappa_2, h\rangle$, thereby neglecting critical subspace structure.

This limitation reveals that, in its conventional formulation, Physics Attention cannot encode directional information inherent in the learned bases, which may be crucial for distinguishing certain physical states. To overcome this deficiency, it is necessary to endow the state tokens with directional awareness of the learnable bases. Concretely, this is achieved by embedding positional information into the state tokens within the functional space $L^2(\Omega)$, thereby allowing the self-attention mechanism to exploit not only projection magnitudes but also geometric orientations. Building on this principle, we propose an enhanced Physics Attention mechanism incorporating positional encoding, which will be detailed in the following sections.

## 2.4 COORDINATE REPRESENTATION OF PHYSICAL STATES

As discussed in Sec. 2.3, Physics Attention is designed to learn a set of robust functional bases $\{\kappa_1, \ldots, \kappa_M\}$ and to compute the projections of the input embeddings $\boldsymbol{Z}$ (i.e., the input function $h$ defined over the domain $\Omega$) onto these bases. A natural and effective way to characterize the relationships among these bases is to represent them in terms of their coordinates with respect to a common set of orthogonal bases, which enables a systematic analysis of their relative configurations.

Similarly, the Physics Cross-Attention mechanism—which performs standard cross-attention between two groups of physical states—can be interpreted from the same projection-based perspective. In this context, it aims to learn two sets of robust functional bases. If the combined set of bases, $\{\kappa_1^q, \ldots, \kappa_{M_q}^q, \kappa_1^k, \ldots, \kappa_{M_k}^k\}$, is defined over the same domain $\Omega$, it can likewise be represented in terms of their coordinates with respect to a common orthogonal basis, allowing for a unified and coherent analysis of their relative configurations. In the following parts, we present some methods for constructing such orthogonal bases and for computing the corresponding coordinate representations.

**Functional Principal Component Analysis (FPCA) Method:** A classic approach for computing these coordinates is to employ spectral decomposition within the Hilbert space framework. To derive the coordinate representations using FPCA(Shang, 2014), we first center the set of basis functions $\{\kappa_1, \ldots, \kappa_M\}$ and define the corresponding empirical covariance operator $\mathcal{R}$ as Eq. (11).

$$
\tilde{\kappa}_i = \kappa_i - \frac{1}{M}\sum_{j=1}^M \kappa_j, \quad i = 1, \ldots, M, \qquad \mathcal{R} = \frac{1}{M}\sum_{j=1}^M \tilde{\kappa}_j \otimes \tilde{\kappa}_j
\tag{11}
$$

The goal of FPCA is then to identify a function $\epsilon \in L^2(\Omega)$ with unit norm, $\|\epsilon\|_{L^2(\Omega)} = 1$, such that it satisfies the eigenvalue problem as follows:

$$
\mathcal{R}\epsilon = \lambda\epsilon, \quad \text{where} \quad \mathcal{R}\epsilon = \frac{1}{M}\sum_{j=1}^M \langle \tilde{\kappa}_j, \epsilon\rangle \tilde{\kappa}_j
\tag{12}
$$

Here, $\mathcal{R}$ represents the empirical covariance operator acting on the centered functional observations $\{\tilde{\kappa}_j\}_{j=1}^M$, and $\lambda$ denotes the eigenvalue associated with the principal component $\epsilon$. Since the operator $\mathcal{R}$ has rank at most $M$, all nonzero eigenfunctions necessarily belong to the subspace $\mathrm{span}\{\tilde{\kappa}_1, \ldots, \tilde{\kappa}_M\}$. Hence, each eigenfunction $\epsilon_i$ can be represented as a linear combination of the centered sample functions defined as follows:

$$\epsilon_i = \boldsymbol{\beta}_i^\top \tilde{\kappa} = \sum_{j=1}^M \beta_{ij} \tilde{\kappa}_j, \quad i = 1, \ldots, M \tag{13}$$

where $\boldsymbol{\beta}_i \in \mathbb{R}^M$ denotes the coefficient vector. Substituting Eq. (12) into the eigenvalue equation $\mathcal{R}\epsilon = \lambda\epsilon$ and introducing the Gram matrix $\boldsymbol{K} \in \mathbb{R}^{M \times M}$ with entries $\boldsymbol{K}_{ij} = \langle \tilde{\kappa}_i, \tilde{\kappa}_j \rangle$, we obtain:

$$\lambda_i \boldsymbol{\beta}_i^\top \tilde{\kappa} = \frac{1}{M} \sum_{j=1}^M \langle \tilde{\kappa}_j, \epsilon_i \rangle \tilde{\kappa}_j = \frac{1}{M} \sum_{j=1}^M \sum_{l=1}^M \beta_{il} \langle \tilde{\kappa}_j, \tilde{\kappa}_l \rangle \tilde{\kappa}_j = \frac{1}{M} (\boldsymbol{K}\boldsymbol{\beta}_i)^\top \tilde{\kappa}, \quad i = 1, \ldots, M \tag{14}$$

By comparing the leftmost and rightmost terms in Eq. (14), we conclude that $\frac{1}{M} \boldsymbol{K} \boldsymbol{\beta}_i = \lambda_i \boldsymbol{\beta}_i$, which implies that $\boldsymbol{\beta}_i$ is an eigenvector of the Gram matrix $\boldsymbol{K}$ corresponding to eigenvalue $M\lambda_i \geq 0$. Consequently, the original infinite-dimensional eigenvalue problem in $L^2(\Omega)$ is reduced to a finite-dimensional eigenvalue problem for the Gram matrix $\boldsymbol{K}$, which can be readily solved using standard linear algebraic techniques.

Let $\boldsymbol{\eta}_i$ denote a unit eigenvector of $\boldsymbol{K}$ corresponding to the eigenvalue $M\lambda_i$. By setting $\boldsymbol{\beta}_i = a\boldsymbol{\eta}_i$ with $a > 0$, the constant $a$ is determined by the normalization condition $|\epsilon_i|_H = 1$. Specifically,

$$\|\epsilon_i\|_H^2 = \boldsymbol{\beta}_i^\top \boldsymbol{K} \boldsymbol{\beta}_i = a^2 \boldsymbol{\eta}_i^\top (M\lambda_i \boldsymbol{\eta}_i) = a^2 M\lambda_i = 1, \quad \Rightarrow \quad a = (M\lambda_i)^{-1/2}. \tag{15}$$

Consequently, the coordinates of the original uncentered functional bases $\{\kappa_1, \ldots, \kappa_M\}$ with respect to the orthogonal eigenbasis $\{\epsilon_1, \ldots, \epsilon_M\}$ are encoded in the matrix $\boldsymbol{\Theta} \in \mathbb{R}^{M \times M}$, defined as

$$\boldsymbol{\Theta}_{ij} = \langle \kappa_i, \epsilon_j \rangle = \frac{(\tilde{\kappa}_i + \bar{\kappa})^\top \tilde{\kappa}^\top \boldsymbol{\eta}_j}{\sqrt{M\lambda_j}} = \frac{\boldsymbol{e}_i \boldsymbol{K} \boldsymbol{\eta}_j}{\sqrt{M\lambda_j}} + C_j = \sqrt{M\lambda_j} \, \boldsymbol{e}_i \boldsymbol{\eta}_j + C_j. \tag{16}$$

where $\boldsymbol{e}_i$ is the $i$-th canonical basis vector with 1 in the $i$-th position and 0 elsewhere and $C_j$ denotes a constant that does not depend on $i$. The computational complexity of this procedure is $O(N + M^3)$. For completeness, we also present two alternative schemes for computing the coordinates. However, as their empirical performance on the benchmarks proved inferior to that of the aforementioned method, we provide their descriptions in Appendix A.2 for reference.

## 2.5 Position-Enhanced Physics Attention with Positional Encoding

The vanilla attention mechanism is inherently position-agnostic when positional information is not explicitly incorporated into the input features. To overcome this limitation, the Galerkin/Fourier Transformer (Cao, 2021) augments each attention head by concatenating spatial coordinates with latent embeddings, and further applies a spectral convolutional decoder (Li et al., 2020) on top of the attention layers. Here, we adopt Rotary Position Embeddings (RoPE) (Su et al., 2021) to encode positional information. Originally proposed for modeling relative positions in language models, RoPE offers a flexible and effective mechanism for integrating coordinate-dependent information into the attention framework.

Following the discussion in Sec. 2.4, we obtain the $M$-dimensional coordinate matrix $\boldsymbol{\Theta} \in \mathbb{R}^{M \times M}$, which represents the $M$ physical states $\boldsymbol{s}_{i=1}^M$. Since $M$ is typically set to 32 or higher, directly embedding them into tokens is often impractical. In addition, not all coordinate components are relevant to the underlying physical interactions among states. To address this, we introduce a learnable linear transformation that projects the $M$-dimensional coordinates into a lower-dimensional latent space as $\boldsymbol{\Theta} \leftarrow \boldsymbol{T}\boldsymbol{\Theta}$. Importantly, the projection matrix $\boldsymbol{T}$ is shared across the entire model.

For a given latent coordinate vector $\theta = \boldsymbol{\Theta}_{i,\cdot}$ and its corresponding $H$-dimensional latent embedding $\boldsymbol{q} = S_{i,\cdot} \boldsymbol{W}_q$, we select the latent dimension $H$ such that $H/(2J) = \gamma \in \mathbb{N}^+$, and accordingly reshape $\boldsymbol{q}$ into a matrix $\tilde{\boldsymbol{q}} \in \mathbb{R}^{J \times 2\gamma}$. Subsequently, we define the RoPE operator $\phi(\text{Coor}, \text{Emb})$ as:

$$\phi(\boldsymbol{q}, \theta) = \mathrm{Concat}(\boldsymbol{R}(\theta_1)\tilde{\boldsymbol{q}}_1, \ldots, \boldsymbol{R}(\theta_J)\tilde{\boldsymbol{q}}_J), \quad \boldsymbol{R}(\theta_j) = \mathrm{diag}(\boldsymbol{R}_1(\theta_j), \ldots, \boldsymbol{R}_\gamma(\theta_j))$$

$$R_k(\theta_j) = \begin{bmatrix} \cos(\theta_j \nu_k) & -\sin(\theta_j \nu_k) \\ \sin(\theta_j \nu_k) & \cos(\theta_j \nu_k) \end{bmatrix} \text{ where } \nu_k = 10000^{-(k-1)/\gamma}, k = 1, \ldots, \gamma \tag{17}$$

where $\mathrm{diag}$ denotes the construction of a block-diagonal matrix by placing the sub-matrices along its diagonal, and the frequency coefficients $\nu_k$ are defined as in (Vaswani et al., 2017; Su et al., 2021). The operator $\Phi(\cdot)$ is then naturally defined by applying the $\phi(\cdot)$ operator row-wise across the entire coordinate matrix.

We summarize our proposed **Position-Enhanced Physics Attention** (PPA) as follows. For **Self-attention** (Self-PPA) and **Cross-attention** (Cross-PPA), Eq. (4) is adapted into Eq. (18) and Eq. (19), respectively, as an extension of the vanilla physics attention(Wu et al., 2024).

$$\boldsymbol{Q}, \boldsymbol{K}, \boldsymbol{V} = \mathrm{Linear}(\boldsymbol{S}), \quad \boldsymbol{\Theta} = \mathrm{Linear}[\mathrm{Get\_Coor}(\boldsymbol{W})],$$

$$\boldsymbol{S}' = \mathrm{Softmax}\left(\frac{\Phi(\boldsymbol{Q}, \boldsymbol{\Theta})\,\Phi(\boldsymbol{K}, \boldsymbol{\Theta})^\top}{\sqrt{C}}\right)\boldsymbol{V} \tag{18}$$

$$\boldsymbol{Q} = \mathrm{Linear}(\boldsymbol{S}_q), \quad \boldsymbol{K}, \boldsymbol{V} = \mathrm{Linear}(\boldsymbol{S}_k), \quad \boldsymbol{\Theta}_q, \boldsymbol{\Theta}_k = \mathrm{Linear}[\mathrm{Get\_Coor}(\boldsymbol{W}_q, \boldsymbol{W}_k)],$$

$$\boldsymbol{S}' = \mathrm{Softmax}\left(\frac{\Phi(\boldsymbol{Q}, \boldsymbol{\Theta}_q)\,\Phi(\boldsymbol{K}, \boldsymbol{\Theta}_k)^\top}{\sqrt{C}}\right)\boldsymbol{V} \tag{19}$$

Here, the procedure $\mathrm{Get\_Coor}$ is defined as in Sec. 2.4.

## 3 MODEL ARCHITECTURE

**Query Encoder** Given a query position $\boldsymbol{y}_i$, the query encoder employs a shared point-wise MLP whose first layer implements a random Fourier feature mapping $\mathcal{Q}(\cdot)$ (Tancik et al., 2020; Rahimi & Recht, 2007). The Gaussian random Fourier mapping is defined as:

$$\mathrm{Query} := \mathcal{Q}(\boldsymbol{Y}) = \mathrm{Concat}\left[\cos\left(2\pi\,\boldsymbol{Y}\boldsymbol{B}\right), \sin\left(2\pi\,\boldsymbol{Y}\boldsymbol{B}\right)\right], \tag{20}$$

where $\boldsymbol{Y} = [\boldsymbol{y}_1, \ldots, \boldsymbol{y}_{n'}]^\top \in \mathbb{R}^{N' \times \dim(\Omega)}$ and $\boldsymbol{y}_i \in \mathbb{R}^d$ denotes the Cartesian coordinates of the $i$-th query point. The projection matrix $\boldsymbol{B} \in \mathbb{R}^{d \times d'}$ has entries sampled i.i.d. from the Gaussian distribution $\mathcal{N}(0, \sigma^2)$, where $\sigma$ is a predefined scale parameter. The concatenation of cosine and sine components yields a $2d'$-dimensional embedding per point. By mapping input coordinates to a trigonometric basis with higher-frequency components, $\mathcal{Q}(\cdot)$ serves to mitigate the spectral bias of coordinate-based neural networks (Tancik et al., 2020; Mildenhall et al., 2020); analogous encodings have also been incorporated into physics-informed machine learning (Wang et al., 2021).

**Design on Heterogeneous Geometry** For operator learning tasks defined on heterogeneous geometries (i.e., when the input and output are discretized differently), we adopt an encoder–decoder backbone based on the canonical transformer architecture (Vaswani et al., 2017), as illustrated in Fig. 1. The detailed implementation is summarized in Eq. (21). We refer to this modified architecture as **HGsolver**. In the formulation below, $n_e$ and $n_d$ denote the depths of the encoder and decoder, respectively, and FF represents the feedforward neural network.

$$\mathrm{KV}^{(l+1)} = \left(\mathrm{FF} \circ \mathrm{Self\text{-}PPA}\right)\left(\mathrm{KV}^{(l)}\right), \qquad l = 0, \ldots, n_e - 1, \quad \mathrm{KV}^{(0)} = \mathrm{Input},$$

$$\mathrm{Out}^{(l+1)} = \left(\mathrm{FF} \circ \mathrm{Cross\text{-}PPA}\right)\left(\mathrm{Out}^{(l)}, \mathrm{KV}^{(n_e)}\right), \quad l = 0, \ldots, n_d - 1, \quad \mathrm{Out}^{(0)} = \mathrm{Query}. \tag{21}$$

In Cross_PPA, we highlight that the query basis $\kappa_i$ and the key basis $\kappa_j$ may be discretized on distinct meshes, $\boldsymbol{X} \in \mathbb{R}^{N \times \dim(\Omega)}$ and $\boldsymbol{Y} \in \mathbb{R}^{N' \times \dim(\Omega)}$, respectively. Under this condition, the inner product $\langle \kappa_i, \kappa_j \rangle$ is computed using a learnable attention matrix $\boldsymbol{L} \in \mathbb{R}^{N \times N'}$, which serves as the kernel matrix for the inner product, as described in Eq. (22). Here, $\boldsymbol{L}$ is shared across the HGsolver, and $\boldsymbol{v}$ represents the slice weights defined in Sec. 2.3.

$$\langle \kappa_i, \kappa_j \rangle = (\boldsymbol{v}_i^{(q)})^\top \boldsymbol{L}(\boldsymbol{v}_j^{(k)}), \quad \boldsymbol{L} = \mathrm{Softmax}\left(\frac{(\boldsymbol{X}\boldsymbol{W}_q^{(\mathrm{pos})})(\boldsymbol{Y}\boldsymbol{W}_k^{(\mathrm{pos})})^\top}{\dim(\Omega)}\right) \tag{22}$$

**Design on Homogeneous Geometry** In operator learning tasks performed on homogeneous geometries (i.e., where the input and output share the same shape and exhibit a one-to-one mapping between corresponding positions), we extend the architecture of Transolver (Wu et al., 2024) by replacing its standard physics attention mechanism with our position-enhanced physics attention. This modified architecture is referred to as **TransolverXP**.

Table 1: Relative $\ell_2$ errors of HGsolver on inverse (**left**), reconstruction (**middle**), and forward (**right**) tasks. *w/o PE* denotes HGsolver with vanilla physics attention. For the Shape-Net car benchmark, the $\ell_2$ errors of the surrounding physical fields are reported.

| Sparse-Ratio | Model | N-S | Elasticity | Darcy | Airfoil | Pipe | ShN-Car |
|:---:|:---:|:---:|:---:|:---:|:---:|:---:|:---:|
| 10% | HG | 0.28 | 0.08 | 0.04 | 0.19 | 0.16 | 0.23 |
| | *w/o PE* | 0.31 | 0.15 | 0.05 | 0.19 | 0.18 | 0.29 |
| 30% | HG | 0.24 | 0.08 | 0.02 | 0.17 | 0.06 | 0.18 |
| | *w/o PE* | 0.24 | 0.14 | 0.02 | 0.18 | 0.09 | 0.22 |
| 50% | HG | 0.19 | 0.07 | 0.01 | 0.10 | 0.04 | 0.16 |
| | *w/o PE* | 0.21 | 0.12 | 0.01 | 0.13 | 0.06 | 0.20 |

## 4 EXPERIMENTS

In this section, we evaluate HGsolver on heterogeneous geometry benchmarks and TransolverXP on homogeneous ones. These benchmarks span different structures and dimensions, and cover forward, inverse, and reconstruction tasks, providing a convincing evaluation of our proposed models.

**Datasets:** Our experiments encompass a broad range of problem settings across both 2D and 3D domains, including point clouds (Elasticity), structured meshes (Plasticity, Airfoil, Pipe), regular grids (Navier–Stokes, Darcy), and unstructured meshes (ShapeNet Car, AirfRANS). The Elasticity, Plasticity, Airfoil, Pipe, Navier–Stokes, and Darcy benchmarks were first introduced in FNO Li et al. (2021) and geo-FNO Li et al. (2022), and have since become widely adopted in subsequent studies. Beyond these standard benchmarks, we further consider design-oriented tasks: the ShapeNet Car dataset Umetani & Bickel (2018), which involves predicting surface pressure and surrounding air velocity from vehicle geometries, and the AirfRANS dataset Bonnet et al. (2022), which provides high-fidelity Reynolds-Averaged Navier–Stokes simulations of airfoils derived from the National Advisory Committee for Aeronautics. In addition, we include an inverse problem benchmark introduced by LNO (Wang & Wang, 2024), involving the solution of the inverse problem for the Burgers equation. Detailed descriptions and specifications of all the aforementioned benchmarks are provided in Appendix A.1.

**Baselines:** We conduct a comprehensive comparison of HGsolver against more than 20 baseline models. These include typical neural operators such as DeepOnet (Lu et al., 2021), FNO (Li et al., 2021), U-NO (Rahman et al., 2023), and LSM (Wu et al., 2023); Transformer-based PDE solvers such as GNOT (Hao et al., 2023) and FactFormer (Li et al., 2023a); as well as classical geometric deep models including PointNet (Qi et al., 2017), GraphSAGE (Hamilton et al., 2017), and Mesh-GraphNet (Pfaff et al., 2021). Among these, LSM (Wu et al., 2023) and GNOT (Hao et al., 2023) represent the previous sota on standard benchmarks, while GINO Li et al. (2023b) and 3D-GeoCA (Deng et al., 2024) are advanced models designed for large-scale, industrial-level simulation benchmarks. In addition, we also compare with recent physics-attention-based models, including Transolver (Wu et al., 2024), LNO (Wang & Wang, 2024), and Transolver++ (Luo et al., 2025).

**Setup:** We conducted our experiments within an open source framework *Neural Solver Library*(Wu et al., 2024). All experiments were run on 8 Nvidia RTX4090 GPUs with 24GB memory.

### 4.1 STANDARD HETEROGENEOUS GEOMETRIES BENCHMARKS

To assess the impact of incorporating positional encoding into physics attention, we design a comprehensive set of PDE tasks encompassing forward, inverse, and reconstruction processes. These tasks provide a rigorous and convincing evaluation of our approach, with detailed implementations presented in Appendix A.1.2. As shown in Tab. 1, physics attention enhanced with the proposed positional encoding demonstrates superior performance across this diverse set of tasks, particularly when handling complex heterogeneous geometries, such as those in the Shape-Net Car and Elasticity benchmarks. Compared to vanilla physics attention, our enhanced approach more effectively captures the intrinsic physical interactions across a wide range of PDE scenarios.

### 4.2 HETEROGENEOUS GEOMETRIES BENCHMARKS ON INVERSE PROBLEM

To ensure a rigorous and fair comparison, we evaluate HGsolver against state-of-the-art models for PDE tasks on heterogeneous geometries, using the inverse problem benchmark of the Burgers equation shown in Appendix. A.1.2. As shown in Tab. 2, the proposed HGsolver consistently demon-

Table 2: Relative MAE of inner reconstructions from *completers* under different observation ratios (**left**), and of outlier reconstructions from *propagators* using eithor ground truth(G.T.) or reconstructed inputs from different observation ratios (**right**).

| Task | Completer | | | | | Propagator | | |
|---|---|---|---|---|---|---|---|---|
| **Models** | 20% | 10% | 5% | 1% | 0.5% | G.T. | 10% | 1% |
| DeepONet | 2.51% | 2.59% | 2.82% | 3.25% | 4.82% | 7.34% | 11.14% | 13.87% |
| GNOT | 1.12% | 1.39% | 1.62% | 1.63% | 2.56% | 5.45% | 8.04% | 9.91% |
| LNO | 0.60% | 0.74% | 0.77% | 1.18% | **2.05**% | 3.73% | 5.69% | 7.72% |
| HGsolver(Ours) | **0.52**% | **0.59**% | **0.67**% | **1.16**% | 2.11% | **3.55**% | **5.61**% | **7.69**% |

Table 3: Performance comparison on design-oriented tasks. Besides the relative $\ell_2$ error of the surrounding (Volume) and surface (Surf) physics fields, the relative $\ell_2$ errors of the drag coefficient ($C_D$) and lift coefficient ($C_L$) are also recorded, along with their Spearman's rank correlations $\rho_D$ and $\rho_L$. For clarity, the best results are in **bold** and complete table is provided in Tab 6.

| MODEL[*] | SHAPE-NET CAR | | | | AIRFRANS | | | |
|---|---|---|---|---|---|---|---|---|
| | VOLUME ↓ | SURF ↓ | $C_D$ ↓ | $\rho_D$ ↑ | VOLUME ↓ | SURF ↓ | $C_L$ ↓ | $\rho_L$ ↑ |
| ... | ... | ... | ... | ... | ... | ... | ... | ... |
| 3D-GEOCA | 0.0319 | 0.0779 | 0.0159 | 0.9842 | / | / | / | / |
| TRANSOLVER | 0.0207 | 0.0745 | 0.0103 | 0.9935 | 0.0037 | 0.0142 | **0.1030** | **0.9978** |
| **TRANSOLVERXP** | **0.0199** | **0.0622** | **0.0095** | **0.9938** | **0.0031** | **0.0094** | 0.1535 | 0.9963 |

strates state-of-the-art performance across varying observation ratios, both in its roles as *propagators* and *completers*, thereby highlighting the decoupling capabilities of HGsolver.

### 4.3 STANDARD HOMOGENEOUS GEOMETRIES BENCHMARKS

We further evaluate **TransolverXP** on eight homogeneous geometry benchmarks. As shown in Tab. 3 and Tab. 5, **TransolverXP** consistently demonstrates competitive performance across these benchmarks, with particularly notable results on the unstructured and complex cases (ShapeNet-Car and AirfRANS), highlighting its effectiveness in handling challenging geometries.

### 4.4 EFFICIENCY ANALYSIS

To evaluate the feasibility and scalability of our proposed method, a critical analysis involves examining the execution time of the eigenvalue problem discussed in Sec. 2.4. Theoretically, the computational complexity of this algorithm is $O(N + M^3)$. In practice, for example in the Airfoil case, the total inference time is 0.55 seconds, of which 0.38(70%) seconds is spent on solving the eigenvalue problem. This corresponds to $M = 64$ and $N = 11,271$. A detailed analysis of execution time of the eigenvalue problem is shown in Appendix A.4. As a side note, since our model introduces only negligible additional parameters, its GPU memory consumption remains virtually the same as that of Transolver(Wu et al., 2024) when the network depth and width are fixed.

## 5 CONCLUSION

**Insights** In this work, we introduced HGsolver, a position-enhanced physics attention framework for modeling PDEs on heterogeneous geometries. Our experimental results highlight the pivotal importance of explicitly capturing interactions among physical states to faithfully represent intrinsic dynamics in heterogeneous domains. Moreover, we present a unified projection-based perspective on physics attention, which not only provides a rigorous theoretical foundation for understanding physical states and the principles underlying their interactions but also offers principled guidance for the development of future neural PDE solvers in physical state spaces. We envisage that the proposed framework, together with its underlying theoretical foundation, will significantly advance the practical resolution of complex PDE problems defined on heterogeneous geometries.

**Limitations and Future Work** A primary limitation of HGsolver is its computational efficiency, as the eigenvalue computation dominates inference time. This bottleneck may be exacerbated in large-scale or high-resolution scenarios. Future work will investigate potential approximations and solver optimizations to alleviate the runtime bottleneck, while aiming to maintain accuracy and generalizability across diverse PDE tasks and heterogeneous geometries.

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

## A  APPENDIX

### A.1  DATASETS SETTINGS

In this section, we provide the details of our experiments, including **benchmarks**, **metrics**, and **implementations**.

#### A.1.1  DESCRIPTIONS OF STANDARD BENCHMARKS

We extensively evaluate our model in eight benchmarks, whose information is summarized in Table 4. Note that these benchmarks involve the following three types of PDEs:

- **Solid material** (Dym et al., 1973): Elasticity and Plasticity.
- **Navier-Stokes equations for fluid** (McLean, 2012): Airfoil, Pipe, Navier-Stokes, Shape-Net Car and AirfRANS.
- **Darcy's law** (Hubbert, 1956): Darcy.

Here are the details of each benchmark.

**Elasticity**  This benchmark is to estimate the inner stress of the elasticity material based on the material structure, which is discretized in 972 points (Li et al., 2022). For each case, the input is a tensor in the shape of $972 \times 2$, which contains the 2D position of each discretized point. The output is the stress of each point, thus in the shape of $972 \times 1$. As for the experiment, 1000 samples with different structures are generated for training and another 200 samples are used for test.

**Plasticity**  This benchmark is to predict the future deformation of the plasticity material under the impact from above by an arbitrary-shaped die (Li et al., 2022). For each case, the input is the shape of the die, which is discretized into the structured mesh and recorded as a tensor with shape $101 \times 31$. The output is the deformation of each mesh point in the future 20 time steps, that is a tensor in the shape of $20 \times 101 \times 31 \times 4$, which contains the deformation in four directions. Experimentally, 900 samples with different die shapes are used for model training and 80 new samples are for test.

**Airfoil**  This task is to estimate the Mach number based on the airfoil shape, where the input shape is discretized into structured mesh with shape $221 \times 51$ and the output is the Mach number for each mesh point (Li et al., 2022). Here, all the shapes are deformed from the NACA-0012 case provided by the National Advisory Committee for Aeronautics. 1000 samples in different airfoil designs are used for training and the other 200 samples are for testing.

**Pipe**  This benchmark is to estimate the horizontal fluid velocity based on the pipe structure (Li et al., 2022). Each case discretizes the pipe into structured mesh with size $129 \times 129$. Thus, for each case, the input tensor is in the shape of $129 \times 129 \times 2$, which contains the position of each discretized mesh point. The output is the velocity value for each point, thus in the shape of $129 \times 129 \times 1$. 1000 samples with different pipe shapes are used for model training and 200 new samples are for test, which are generated by controlling the centerline of the pipe.

Table 4: Summary of experiment benchmarks, where the first six datasets are from FNO (Li et al., 2021) and geo-FNO (Li et al., 2022), Shape-Net Car is from (Umetani & Bickel, 2018) and preprocessed by (Deng et al., 2024), and AirfRANS is from (Bonnet et al., 2022). #Mesh records the size of discretized meshes. #Dataset is organized as the number of samples in training and test sets.

| GEOMETRY | BENCHMARKS | #DIM | #MESH | #INPUT | #OUTPUT | #DATASET |
|---|---|---|---|---|---|---|
| POINT CLOUD | ELASTICITY | 2D | 972 | STRUCTURE | INNER STRESS | (1000, 200) |
| STRUCTURED MESH | PLASTICITY AIRFOIL PIPE | 2D+TIME 2D 2D | 3,131 11,271 16,641 | EXTERNAL FORCE STRUCTURE STRUCTURE | MESH DISPLACEMENT MACH NUMBER FLUID VELOCITY | (900, 80) (1000, 200) (1000, 200) |
| REGULAR GRID | NAVIER–STOKES DARCY | 2D+TIME 2D | 4,096 7,225 | PAST VELOCITY POROUS MEDIUM | FUTURE VELOCITY FLUID PRESSURE | (1000, 200) (1000, 200) |
| UNSTRUCTURED MESH | SHAPE-NET CAR AIRFRANS | 3D 2D | 32,186 32,000 | STRUCTURE STRUCTURE | VELOCITY & PRESSURE VELOCITY & PRESSURE | (789, 100) (800, 200) |

**Navier-Stokes**   This benchmark is to model the incompressible and viscous flow on a unit torus, where the fluid density is constant and viscosity is set as $10^{-5}$ (Li et al., 2021). The fluid field is discretized into $64 \times 64$ regular grid. The task is to predict the fluid in the next 10 steps based on the observations in the past 10 steps. 1000 fluids with different initial conditions are generated for training, and 200 new samples are used for test.

**Darcy**   This benchmark is to model the flow of fluid through a porous medium (Li et al., 2021). Experimentally, the process is discretized into a $421 \times 421$ regular grid. Then we downsample the data into $85 \times 85$ resolution for main experiments. The input of the model is the structure of the porous medium and the output is the fluid pressure for each grid. 1000 samples are used for training and 200 samples are generated for test, where different cases contain different medium structures.

**Shape-Net Car**   This benchmark focuses on the drag coefficient estimation for the driving car, which is essential for car design. Overall, 889 samples with different car shapes are generated to simulate the 72 km/h speed driving situation (Umetani & Bickel, 2018), where the car shapes are from the "car" category of ShapeNet (Chang et al., 2015). Concretely, they discretize the whole space into unstructured mesh with 32,186 mesh points and record the air around the car and the pressure over the surface. Here we follow the experiment setting in 3D-GeoCA (Deng et al., 2024), which takes 789 samples for training and the other 100 samples for testing. The input mesh of each sample is also preprocessed into the combination of mesh point position, signed distance function and normal vector. The model is trained to predict the velocity and pressure value for each point. Afterward, we can calculate the drag coefficient based on these estimated physics fields.

**AirfRANS**   This dataset contains the high-fidelity simulation data for Reynolds-Averaged Navier–Stokes equations (Bonnet et al., 2022), which is also used to assist airfoil design. Different from Airfoil (Li et al., 2022), this benchmark involves more diverse airfoil shapes under finer discretized meshes. Specifically, it adopts airfoils in the 4 and 5 digits series of the National Advisory Committee for Aeronautics, which have been widely used historically. Each case is discretized into 32,000 mesh points. By changing the airfoil shape, Reynolds number, and angle of attack, AirfRANS provides 1000 samples, where 800 samples are used for training and 200 for the test set. Air velocity, pressure and viscosity are recorded for surrounding space and pressure is recorded for the surface. Note that both drag and lift coefficients can be calculated based on these physics quantities. However, as their original paper stated, air velocity is hard to estimate for airplanes, making all the deep models fail in drag coefficient estimation (Bonnet et al., 2022). Thus, in the main text, we focus on the lift coefficient estimation and the pressure quantity on the volume and surface, which is essential to the take-off and landing stages of airplanes.

### A.1.2   IMPLEMENTATIONS OF PDE TASKS

**Navier-Stokes (inverse)**   This inverse task aims to infer the initial fluid state from partially observed future states. The input consists of velocity and pressure fields at a subset of grid points that are masked over the next 10 time steps on a $64 \times 64$ regular grid. The output corresponds to the complete initial fluid state at full spatial resolution. This inverse task evaluates the model's ability to reconstruct physically consistent initial conditions from limited, masked temporal observations.

**Elasticity (reconstruction)**   This reconstruction task aims to predict the complete stress field from partially observed stress values. The input consists of stress values at unmasked points of the $972 \times 1$ discretized grid, and the output corresponds to the stress values at the masked locations. This task evaluates the model's ability to infer the full stress distribution from partially observed measurements.

**Darcy (reconstruction)**   This reconstruction task involves estimating the fluid pressure at masked grid points given partially observed pressures. The input consists of pressures at unmasked points on the downsampled $421 \times 421$ grid, and the output corresponds to the pressure values at the masked locations. This task assesses the model's capability to reconstruct the complete pressure field in heterogeneous porous media from limited observations.

**Pipe (forward)**   This forward task aims to predict the horizontal fluid velocity from the pipe geometry. The input consists of the spatial coordinates of all $129 \times 129$ mesh points, with a subset masked according to a predefined ratio, and the output corresponds to the velocity at each point, represented

as a tensor of shape $129 \times 129 \times 1$. This task evaluates the model's ability to map partially observed geometric configurations to the resulting fluid dynamics.

**Airfoil (forward)**  This forward task aims to predict the Mach number distribution based on the discretized airfoil shape. The input consists of coordinates of a subset of the $221 \times 51$ structured mesh, masked according to a predefined ratio, and the output corresponds to the Mach number at each mesh point. This task examines the model's capability to infer aerodynamic properties from partially observed geometric features.

**Shape-Net Car (forward)**  This forward task aims to predict the velocity and pressure fields around a car from its 3D geometry. The input consists of a preprocessed representation of 32,186 mesh points, including positions, signed distance functions, and normal vectors, with a subset masked according to a predefined ratio. The output corresponds to the velocity and pressure values for all points. This task evaluates the model's effectiveness in estimating complex fluid interactions from partially observed 3D vehicle geometries under simulated driving conditions.

### A.1.3   INVERSE PROBLEM BENCHMARKS

Here, we will introduce an inverse benchmark from LNO(Wang & Wang, 2024) as below.

To demonstrate the flexibility of the model, an inverse problem is designed. Given a partially observed solution $u(x)$, the objective is to recover the complete solution $u(x)$ over a larger domain. Specifically, the test is conducted on the 1D Burgers' equation:

$$\frac{\partial}{\partial t}u(x,t) = 0.01 \ \frac{\partial^2}{\partial x^2}u(x,t) - u(x,t)\frac{\partial}{\partial x}u(x,t), \quad x \in [0,1], \ t \in [0,1]$$

$$u(x,0) \sim \text{GaussianProcess}(0, \exp\big(-\tfrac{2}{pl^2}\sin^2(\pi||x-x^{'}||)\big)), \quad u(0,t) = u(1,t)$$

The ground-truth data is generated on a $128 \times 128$ grid with periodic boundary conditions. Initial conditions are sampled from a Gaussian process with periodic length $p = 1$ and scaling factor $l = 1$.

The goal of this inverse problem is to reconstruct the complete solution $u(x)$ across the entire spatiotemporal domain $(x,t) \in [0,1] \times [0,1]$, based on sparsely random-sampled or fixed-sampled observations in the subdomain $(x,t) \in [0,1] \times [0.25, 0.75]$.

Instead of using a naive approach that directly learns the mapping from partially observed samples in the subdomain to the complete solution in the whole domain, a two-stage strategy is proposed, inspired by inpainting(Pathak et al., 2016; Yu et al., 2018) and outpainting(Yang et al., 2019; Sabini & Rusak, 2018). First, the model is trained as a completer to interpolate the sparsely sampled points in the subdomain $[0,1] \times [0.25, 0.75]$ to predict all densely and regularly sampled points in the same subdomain. Then, the model is trained as a propagator to extrapolate the results of the completer from the subdomain to the whole domain $[0,1] \times [0,1]$. Since the observation and prediction samples are located in different positions, only models with decoupling properties can be used as the completer and propagator.

The performance of the model is compared with that of DeepONet(Lu et al., 2021), GNOT(Hao et al., 2023) and LNO(Wang & Wang, 2024) in both stages.

### A.1.4   METRICS

Since our experiment consists of standard benchmarks and practical design tasks, we also include several design-oriented metrics in addition to the relative L2 for physics fields.

**Relative L2 for physics fields**  Given the physics field $\mathbf{u}$ and the model predicted field $\widehat{\mathbf{u}}$, the relative L2 of model prediction can be calculated as follows:

$$\text{Relative L2} = \frac{\|\mathbf{u} - \widehat{\mathbf{u}}\|}{\|\mathbf{u}\|}. \tag{23}$$

**Mean Absolute Error (MAE)**   Given the physics field $\mathbf{u}$ and the model predicted field $\widehat{\mathbf{u}}$, the mean absolute error of the model prediction can be calculated as follows:

$$\text{MAE} = \frac{1}{N} \sum_{i=1}^{N} |u_i - \widehat{u}_i|, \tag{24}$$

**Relative L2 for drag and lift coefficients**   For Shape-Net Car and AirFRANS, we also calculated the drag and lift coefficients based on the estimated physics fields. For unit density fluid, the coefficient (drag or lift) is defined as follows:

$$C = \frac{2}{v^2 A} \left( \int_{\partial \Omega} p(\boldsymbol{\xi}) \left( \widehat{n}(\boldsymbol{\xi}) \cdot \widehat{i}(\boldsymbol{\xi}) \right) \mathrm{d}\boldsymbol{\xi} + \int_{\partial \Omega} \tau(\boldsymbol{\xi}) \cdot \widehat{i}(\boldsymbol{\xi}) \mathrm{d}\boldsymbol{\xi} \right), \tag{25}$$

where $v$ is the speed of the inlet flow, $A$ is the reference area, $\partial \Omega$ is the object surface, $p$ denotes the pressure function, $\widehat{n}$ means the outward unit normal vector of the surface, $\widehat{i}$ is the direction of the inlet flow and $\tau$ denotes wall shear stress on the surface. $\tau$ can be calculated from the air velocity near the surface (McCormick, 1994), which is usually much smaller than the pressure item. Specifically, for the drag coefficient of Shape-Net Car, $\widehat{i}$ is set as $(-1, 0, 0)$ and $A$ is the area of the smallest rectangle enclosing the front of cars. As for the lift coefficient of AirFRANS, $\widehat{i}$ is set as $(0, 0, -1)$. The relative L2 is defined between the ground truth coefficient and the coefficient calculated from the predicted velocity and pressure field.

**Spearman's rank correlations for drag and lift coefficients**   Given $K$ samples in the test set with the ground truth coefficients $C = \{C^1, \cdots, C^K\}$ (drag or lift) and the model predicted coefficients $\widehat{C} = \{\widehat{C}^1, \cdots, \widehat{C}^K\}$, the Spearman correlation coefficient is defined as the Pearson correlation coefficient between the rank variables, that is:

$$\rho = \frac{\text{cov}\left( R(C) R(\widehat{C}) \right)}{\sigma_{R(C)} \sigma_{R(\widehat{C})}}, \tag{26}$$

where $R$ is the ranking function, cov denotes the covariance and $\sigma$ represents the standard deviation of the rank variables. Thus, this metric is highly correlated to the model guide for design optimization. A higher correlation value indicates that it is easier to find the best design following the model-predicted coefficients (Spearman, 1961).

## A.2   ADDITIONAL METHOD FOR COORDINATES REPRESENTATION

**Gram-Schmidt Method:**   One straightforward method for computing these coordinates is to apply the Gram-Schmidt orthogonalization procedure (Björck, 1994) to the original bases, as illustrated in Eq. (28). Specifically, given a set of bases $\{\kappa_1, \ldots, \kappa_M\}$, the associated orthogonal bases $\{\epsilon_1, \ldots, \epsilon_M\}$ can be obtained as follows.

$$\begin{aligned} \epsilon_1 &= \frac{\kappa_1}{\|\kappa_1\|}, \quad \|\kappa_1\| = \sqrt{\langle \kappa_1, \kappa_1 \rangle}, \\ \tilde{\epsilon}_i &= \kappa_i - \sum_{j=1}^{i-1} \langle \kappa_i, \epsilon_j \rangle \epsilon_j, \quad \epsilon_i = \frac{\tilde{\epsilon}_i}{\sqrt{\langle \tilde{\epsilon}_i, \tilde{\epsilon}_i \rangle}}, \quad i = 2, \ldots, M \end{aligned} \tag{27}$$

Consequently, the coordinates of the original bases $\kappa$ with respect to the orthogonal bases $\epsilon$ can be represented by the matrix $\boldsymbol{\Theta} \in \mathbb{R}^{M \times M}$, where

$$\boldsymbol{\Theta}_{i,j} = \langle \kappa_i, \epsilon_j \rangle \approx \frac{1}{N} \sum_{k=1}^{N} \kappa_i(x_k) \epsilon_j(x_k), \quad i, j = 1, \ldots, M. \tag{28}$$

Each row $\boldsymbol{\Theta}_i$ can thus be interpreted as the "position" of the state $s_i$ in the space defined by the orthogonal bases. The computational complexity of this procedure is $O(NM^2)$, which may become prohibitively expensive when $M$ is large.

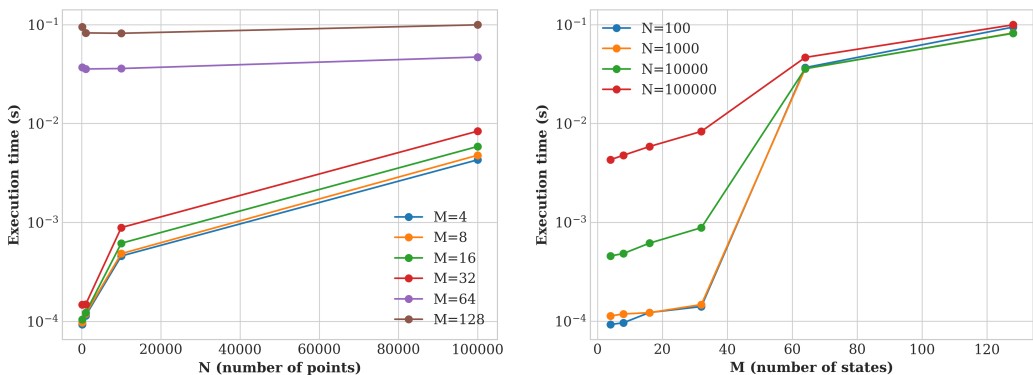

Figure 2: Times Efficency Analysis on Number of States and Points

**Static Bases Method:** Although the aforementioned methods provide a principled approach for computing the coordinates, empirical strategies are also worth considering, as they typically require fewer computational resources. In this approach, a set of bases $\{\epsilon_1, \ldots, \epsilon_E\}$ is selected empirically to compute the coordinates $\boldsymbol{\Theta}$ of $\{\kappa_1, \ldots, \kappa_M\}$ as follows:

$$\boldsymbol{\Theta}_{ij} = \langle \kappa_i, \epsilon_j \rangle \in \mathbb{R}^{M \times E}. \tag{29}$$

Subsequently, the columns of $\boldsymbol{\Theta}$ with the top-$k$ variances are retained and taken as the coordinates of the state tokens. For instance, one may employ a series of Fourier basis functions, as their coefficients can be efficiently computed via the Fast Fourier Transform (FFT). This approach enables the evaluation of the coordinates of $N$ candidate bases with respect to $M$ functions at an overall computational cost of $O(MN \log N)$.

### A.3 ADDITIONAL EXPERIMENTAL RESULTS

The results of **TransolverXP** on standard PDE benchmark is shown in Tab. 5.

### A.4 TIMES EFFICENCY ANALYSIS

The computational complexity of the Gram eigenvalue algorithm is $O(N + M^3)$, where $N$ denotes the number of samples and $M$ represents the feature dimension. The empirical results show that when $M$ is relatively small, the execution time grows markedly with increasing $N$, suggesting that the linear term in the complexity dominates. As $M$ becomes larger, however, the cubic term $M^3$ emerges as the principal factor, leading to a significant increase in runtime while the dependence on $N$ becomes negligible. This observation is consistent with the theoretical analysis and underscores the scalability limitation of the algorithm with respect to high-dimensional features.

### A.5 INPLEMENTATION SETTINGS

Table 7 summarizes the training and model configurations of TransolverXP and HGsolver. Training settings, including learning rate, optimizer, batch size, and number of epochs, are adopted directly from previous works (Wu et al., 2024; Bonnet et al., 2022; Hao et al., 2023; Deng et al., 2024) without additional tuning. Loss functions are defined separately for volume ($\mathcal{L}_{\mathrm{v}}$) and surface fields ($\mathcal{L}_{\mathrm{s}}$), with Darcy additionally incorporating a spatial gradient regularization term ($\mathcal{L}_{\mathrm{g}}$) following ONO (Xiao et al., 2024). For HGsolver, we employ a 4+4 encoder-decoder structure, which is applied consistently across all relevant benchmarks.

### A.6 COMPUTER SYSTEM INFORMATION

Deatailed information of computer system we conduct our experiments are shown in the *sysinfo.txt* of supplementary materials.

Table 5: Relative $\ell_2$ error on standard benchmarks are presented as mean of 4 runs. "/" means that the baseline cannot apply to this benchmark. For clarity, the **first**, **second**, and **third** best are highlighted. All metrics are derived from Transolve++ (Luo et al., 2025), with models maintaining an equivalent number of parameters.

| MODEL | POINT CLOUD ELASTICITY | STRUCTURED MESH PLASTICITY | AIRFOIL | PIPE | REGULAR GRID NAVIER–STOKES | DARCY |
|---|---|---|---|---|---|---|
| FNO | / | / | / | / | 0.1556 | 0.0108 |
| WMT | 0.0359 | 0.0076 | 0.0075 | 0.0077 | 0.1541 | 0.0082 |
| U-FNO | 0.0239 | 0.0039 | 0.0269 | 0.0056 | 0.2231 | 0.0183 |
| geo-FNO | 0.0229 | 0.0074 | 0.0138 | 0.0067 | 0.1556 | 0.0108 |
| U-NO | 0.0258 | 0.0034 | 0.0078 | 0.0100 | 0.1713 | 0.0113 |
| F-FNO | 0.0263 | 0.0047 | 0.0078 | 0.0070 | 0.2322 | 0.0077 |
| LSM | 0.0218 | 0.0025 | 0.0059 | 0.0050 | 0.1535 | 0.0065 |
| GALERKIN | 0.0240 | 0.0120 | 0.0118 | 0.0098 | 0.1401 | 0.0084 |
| HT-NET | / | 0.0333 | 0.0065 | 0.0059 | 0.1847 | 0.0079 |
| OFORMER | 0.0183 | 0.0017 | 0.0183 | 0.0168 | 0.1705 | 0.0124 |
| GNOT | 0.0086 | 0.0336 | 0.0076 | 0.0047 | 0.1380 | 0.0105 |
| FACTFORMER | / | 0.0312 | 0.0071 | 0.0060 | 0.1214 | 0.0109 |
| ONO | 0.0118 | 0.0048 | 0.0061 | 0.0052 | 0.1195 | 0.0076 |
| TRANSOLVER | 0.0064 | 0.0012 | 0.0053 | 0.0033 | 0.0900 | 0.0057 |
| LNO | 0.0069 | 0.0029 | 0.0053 | 0.0031 | 0.0830 | 0.0063 |
| TRANSOLVER++ | 0.0052 | 0.0011 | 0.0048 | 0.0027 | 0.0719 | 0.0049 |
| TRANSOLVERXP | 0.0050 | 0.0015 | 0.0061 | 0.0030 | 0.0842 | 0.0069 |

Table 6: Performance comparison on design-oriented tasks is conducted. In addition to the relative $\ell_2$ error of the surrounding (Volume) and surface (Surf) physics fields, the relative $\ell_2$ errors of the drag coefficient ($C_D$) and lift coefficient ($C_L$) are also recorded, along with their corresponding Spearman's rank correlations $\rho_D$ and $\rho_L$. A Spearman's correlation value close to 1 indicates superior performance. For clarity, the best results are in **bold**.

| MODEL[*] | SHAPE-NET CAR VOLUME ↓ | SURF ↓ | $C_D$ ↓ | $\rho_D$ ↑ | AIRFRANS VOLUME ↓ | SURF ↓ | $C_L$ ↓ | $\rho_L$ ↑ |
|---|---|---|---|---|---|---|---|---|
| SIMPLE MLP | 0.0512 | 0.1304 | 0.0307 | 0.9496 | 0.0081 | 0.0200 | 0.2108 | 0.9932 |
| GRAPHSAGE | 0.0461 | 0.1050 | 0.0270 | 0.9695 | 0.0087 | 0.0184 | 0.1476 | 0.9964 |
| POINTNET | 0.0494 | 0.1104 | 0.0298 | 0.9583 | 0.0253 | 0.0996 | 0.1973 | 0.9919 |
| GRAPH U-NET | 0.0471 | 0.1102 | 0.0226 | 0.9725 | 0.0076 | 0.0144 | 0.1677 | 0.9949 |
| MESHGRAPHNET | 0.0354 | 0.0781 | 0.0168 | 0.9840 | 0.0214 | 0.0387 | 0.2252 | 0.9945 |
| GNO | 0.0383 | 0.0815 | 0.0172 | 0.9834 | 0.0269 | 0.0405 | 0.2016 | 0.9938 |
| GALERKIN | 0.0339 | 0.0878 | 0.0179 | 0.9764 | 0.0074 | 0.0159 | 0.2336 | 0.9951 |
| GNOT | 0.0329 | 0.0798 | 0.0178 | 0.9833 | 0.0049 | 0.0152 | 0.1992 | 0.9942 |
| GINO | 0.0386 | 0.0810 | 0.0184 | 0.9826 | 0.0297 | 0.0482 | 0.1821 | 0.9958 |
| 3D-GEOCA | 0.0319 | 0.0779 | 0.0159 | 0.9842 | / | / | / | / |
| TRANSOLVER | 0.0207 | 0.0745 | 0.0103 | 0.9935 | 0.0037 | 0.0142 | **0.1030** | **0.9978** |
| **TRANSOLVERXP** | **0.0199** | **0.0622** | **0.0095** | **0.9938** | **0.0031** | **0.0094** | 0.1535 | 0.9963 |

Table 7: Training and model configurations of TransolverXP and HGsolver. Training configurations are directly from previous works without extra tuning (Wu et al., 2024; Bonnet et al., 2022; Hao et al., 2023; Deng et al., 2024). Here $\mathcal{L}_v$ and $\mathcal{L}_s$ represent the loss on volume and surface fields respectively. As for Darcy, we adopt an additional spatial gradient regularization term $\mathcal{L}_g$ following ONO (Xiao et al., 2024).

**Here, we emphasize we apply a 4+4 encoder-decoder structure in HGsolver.**

| BENCHMARKS | TRAINING CONFIGURATION (SHARED IN ALL BASELINES) | | | | | MODEL CONFIGURATION | | | |
| --- | --- | --- | --- | --- | --- | --- | --- | --- | --- |
| | LOSS | EPOCHS | INITIAL LR | OPTIMIZER | BATCH SIZE | LAYERS $L$ | HEADS | CHANNELS $C$ | SLICES $M$ |
| ELASTICITY | | | | | 1 | | | 128 | 64 |
| PLASTICITY | | | | | 8 | | | 128 | 64 |
| AIRFOIL | RELATIVE | 500 | $10^{-3}$ | ADAMW | 4 | 8 | 8 | 128 | 64 |
| PIPE | L2 | | | (2019) | 4 | | | 128 | 64 |
| NAVIER–STOKES | | | | | 2 | | | 256 | 32 |
| DARCY | $\mathcal{L}_{rL2} + 0.1\mathcal{L}_g$ | | | | 4 | | | 128 | 64 |
| SHAPE-NET CAR | $\mathcal{L}_v + 0.5\mathcal{L}_s$ | 200 | $10^{-3}$ | ADAM | 1 | 8 | 8 | 256 | 32 |
| AIRFRANS | $\mathcal{L}_v + \mathcal{L}_s$ | 400 | | (2015) | 1 | | | 256 | 32 |

