# OpenReview forum: "HGsolver: Position-Enhanced Physics Attention Informed Heterogeneous Geometries Neural Solver for PDEs"
_ICLR.cc/2026/Conference — ICLR 2026 Conference Withdrawn Submission_

### Official Review · Reviewer_7M5A · 2025-10-27

**Soundness:** 2
**Presentation:** 1
**Contribution:** 2
**Rating:** 2
**Confidence:** 5

**Summary:**

This paper presents HGSolver to tackle PDEs with heterogeneous geometries, where the input and output have distinct geometries. Specifically, HGSolver enhances the Physics Attention mechanism proposed by Transolver by introducing “position information”, which is named as position-enhanced physics attention. Besides, an encoder-decoder framework is also presented to tackle the heterogeneous geometries, in which the decoder processes query geometries by a cross-version of position-enhanced physics attention. As a result, HGSolver can handle sparse observations and inverse problems. The position-enhanced physics attention is better than vanilla physics attention, but is over 2x slower than the vanilla version.

**Strengths:**

(1)	In this paper, the authors interpret that the slice weight distribution contains “position” information of physics tokens. And it is reasonable to introduce “position” information into Physics-Attention.

(2)	The encoder-decoder framework can empower the model to process heterogeneous geometries.

**Weaknesses:**

### (1) Vague motivation and missing baselines.

The motivation for introducing “position” information into physics tokens is reasonable for improving physics learning. However, it is not designed for heterogeneous geometries. Actually, vanilla physics attention with the encoder-decoder architecture can also tackle heterogeneous geometries, which is also an important baseline.

### (2) The efficiency is too bad.

Although the authors have mentioned this issue in the limitations section, I do not think this issue should be neglected. It is worth noticing that vanilla Transolver demonstrates favorable scalability. If we align the running time of Transolver and TransolverXP, it may allow us to add layers to Transolver, which can be better than TransolverXP. Also, the GPU memory is not included.

### (3) Missing visualization of position information.

It is reasonable to introduce the distribution information of slice weight into physics attention. However, there are no intuitive visualizations for the final learned position embedding. I think this is an essential analysis, while the current version does not contain this experiment.

### (4) The current presentation is hard to understand and some statements are without support.

-	For all the citations, there should be a blank between two words, such as “LNO \cite{lno}”.

-	In Section 2.2, the introduction of Physics Attention is too long. Why introduce Transolver++? Does HGSolver employ the physics attention with adaptive temperature in Transolver++?

-	In Section 2.3, there are confusing and inconsistent usages in Z or \mathbf{Z}, and v or \mathbf{v}. Besides, in Section 2.3, no citations for Transolver.

-	In Lines 239-240, the authors claim that “Physics Attention cannot encode directional information inherent in the learned bases, which may be crucial for distinguishing certain physical states.” However, this claim is without any evidence in the experiment section.

-	In Section 2.4, there are too many notations. I think the authors should include the shape of each symbol.

-	In Line 317 of Section 2.5, I would suggest not to reuse the symbol \mathbf{\Theta}. In Line 320, the definition of the RoPE operator is wrong, which should be \phi(Embed, Coor).

-	In Eq. (18)-(19), I think you should introduce these two equations separately.

**Questions:**

What do you mean by “completers” and “propagators” in Table 2.

**Details Of Ethics Concerns:**

This paper has an ethical issue in the appendix A1.1 and A1.4, which are directly copied from Transolver’s appendix B.1 and B.2 without any modification. I think this is a very serious academic ethical issue. The authors should delete them and directly cite the previous paper.

---

### Official Review · Reviewer_sr2Z · 2025-10-29

**Soundness:** 3
**Presentation:** 3
**Contribution:** 3
**Rating:** 6
**Confidence:** 3

**Summary:**

This paper proposes HGsolver, a neural operator framework for PDE tasks on heterogeneous geometries. The main technical contribution is a projection-based view of physics attention: physical states are treated as projections of input embeddings onto learnable functional bases in a Hilbert space. The authors compute coordinate representations of these bases, embed those coordinates into state tokens using a Rotary Position Embedding style operator, and introduce Position-Enhanced Physics Attention for both self- and cross-attention. HGsolver uses an encoder–decoder backbone to handle mismatched input/output discretizations, while a minimally modified variant, TransolverXP, applies PPA to homogeneous settings.

**Strengths:**

1. The projection-in-Hilbert-space interpretation provides a clear and principled theoretical foundation for physics-attention, offering a strong motivation for introducing coordinate-based representations.

2. The paper presents comprehensive experiments across forward, inverse, and reconstruction tasks on diverse 2D/3D and structured/unstructured benchmarks, consistently showing competitive or state-of-the-art results.

3. The encoder–decoder architecture and the lightweight TransolverXP variant demonstrate the generality of the proposed method, making it applicable to both heterogeneous and homogeneous geometry settings with minimal architectural modification.

**Weaknesses:**

1. The paper briefly mentions that FPCA-based coordinate computation outperforms empirical or FFT-based alternatives but lacks quantitative evidence. Could the authors provide a detailed comparison (e.g., accuracy vs. runtime or complexity curves) between FPCA and its faster approximations to substantiate the claimed efficiency-performance balance?

2. It's unclear how well the learned bases generalize when test geometries are qualitatively different from training (e.g., very different mesh densities or boundary conditions). More experiments or discussion would help.

3. The authors claim negligible extra parameters compared to Transolver for fixed depth/width. Could the authors provide more explicit memory/time scaling plots for varying M and N?

**Questions:**

Please see Weaknesses above.

---

### Official Review · Reviewer_UP3y · 2025-10-31

**Soundness:** 2
**Presentation:** 2
**Contribution:** 2
**Rating:** 2
**Confidence:** 5

**Summary:**

This paper extend the Transolver [1] to handle different meshes (heterogeneous geometries) through a Position-Enhanced Physics-Attention module, which is integrated into the vallina tranformer encoder-decoder architechture. For homogeneous cases, as in [1], the Transolver equipped with Position-Enhanced Physics-Attention demonstrates improved performance over the original Transolver with Physics-Attention on ELASTICITY, PIPE, Incompressible Navier-Stokes and SHAPE-NET CAR. However, a performance drop is reported on DARCY, PLASTICITY, AIRFOIL, and certain quantities in AIRFRANS.

**By assuming a "continuous" form of the binary matrix multiplication decomposition, expressed as "slices = weights $\times$ embedded input singals" [1] in Hilbert space, this paper interprets the linear attention in [1] as a weighted inner product between the "row-functions" of the weight matrix in the multiplication.** To obtain a coordinate representation of these "row-functions", this paper employs a generalized. version of Principal Component Analysis (PCA) in Hilbert space [2]. Finally, a standard rotary positional encoding is applied to the vanilla attention mechanism [3, 4].

*[1] Wu, H., Luo, H., Wang, H., Wang, J., & Long, M. (2024, July). Transolver: a fast transformer solver for PDEs on general geometries. In *Proceedings of the 41st International Conference on Machine Learning* (pp. 53681-53705).*

*[2] Shang, H. L. (2014). A survey of functional principal component analysis. *AStA Advances in Statistical Analysis*, *98*(2), 121-142.*

*[3] Vaswani, A., Shazeer, N., Parmar, N., Uszkoreit, J., Jones, L., Gomez, A. N., ... & Polosukhin, I. (2017). Attention is all you need. *Advances in neural information processing systems*, *30*.*

*[4] Su, J., Ahmed, M., Lu, Y., Pan, S., Bo, W., & Liu, Y. (2024). Roformer: Enhanced transformer with rotary position embedding. *Neurocomputing*, *568*, 127063.*

**Strengths:**

- The paper adopts an encoder–decoder architecture based on the vanilla Transformer to handle the differing query structures between inputs and outputs. Specifically, incremental designs on the aggregation in Transolver to adapt to different queries between inputs and outputs is valuable.

- The overall presentation of the paper, from the introduction through the methodology, is clear and well organized.

**Weaknesses:**

- The authors provide no code implementation, which undermines confidence in the reproducibility of their reported results and the claimed efficiency of the model.

- Because no code is provided, I suspect that FPCA (functional PCA) must be solved at each forward pass:

   - If FPCA needs to be computed during every training or inference forward step, this could impose a significant computational burden. Adding such cost is problematic for a model whose main advantage is fast simulation (as claimed for Transolver), unless the accuracy or utility gain is substantial.

   - The purported “benefit” of FPCA seems to be that an external procedure finds a basis well-ordered in the energy sense, which helps reduce MSE.

   - Although Section 2.5 attempts to exploit sparsity of the $T$ matrix to reduce attention cost (and thereby token-processing burden), I believe the primary additional overhead relative to Transolver remains the per-forward FPCA computation (during both training and testing), rather than the speedups from sharing $T$ or the $L$ term in cross_PPA.

   - I also have reservations about the computational complexity analysis provided; see details in the “Questions” section below.

- The model performs poorly on certain benchmarks, such as Darcy, which involve discontinuous fields.

- For additional concerns, see the “Questions” section.

**Questions:**

- To maintain predictive accuracy comparable to Transolver, does the introduction of the additional “direction” increase the size of the attention matrix?

- Why would distinguishing the “direction” be beneficial?
   - It seems plausible that multiple $\kappa$ values can map to the same physical state (helpful for compressing “similar-behavior” information). In the original setting, the same $\langle \kappa, h \rangle$ pair does not need to be distinguished. Under this condition, the network only needs to identify which pair contributes to predicting the final output function.
   - With the added directional information, the network may require more effort (e.g., memory and training epochs) to identify $\langle \kappa, h \rangle$. What is the specific benefit of introducing this inductive bias? How does the newly introduced “directional information“ improve prediction? The paper only briefly mentions this on line 234; can the authors provide further explanation, ideally supported by examples or theoretical justification? Otherwise, the improvement might be attributed to the rotational encoding itself, rather than a problem-specific advantage for operator learning.

- How is this “direction” formally defined? For example, is it a specific type of normalized inner product?

- Since $\kappa$ and $h$ appear symmetric in Eq. 8, why not project $h$ instead, as it naturally integrates information from the input functions? Would the results be the same, making the choice arbitrary?

- Have you considered applying similar random Fourier features to the input functions? If not, why restrict them to the output solution? How does this mitigate spectral bias, and is this theoretically motivated or mainly an empirical benefit of the rotary positional embedding design?

- How is the claimed computational complexity $O(N + M^3)$ derived? If building the Gram matrix is considered, there should be a factor of $M^2$ in front of $N$, yielding $O(M^2 N + M^3)$. For an architecture with $L$ Transolver blocks, including the extra cost of the eigenvalue problem, is the total complexity on homogeneous tasks $O(L M^2 N + M^3)$? If so, this implies a substantial computational burden in practice, especially for the given $M = 2^5$ or $2^6$ configuration. How does this scale for heterogeneous geometries?

Overall, it appears that the proposed method may only provide partial improvements at a high computational cost. Combined with the absence of available code, the reproducibility and practical utility of the model are questionable.

---

### Official Review · Reviewer_hicZ · 2025-11-01

**Soundness:** 2
**Presentation:** 3
**Contribution:** 2
**Rating:** 4
**Confidence:** 4

**Summary:**

This paper primarily introduces HGSolver, a neural operator improved upon the foundation of Transolver, designed for solving partial differential equations (PDEs) with heterogeneous geometries. It views physical states as projections onto learnable function bases in a Hilbert space and, based on this, develops a position-enhanced physics attention mechanism. By integrating coordinate representations through rotary positional embeddings, it enhances the modeling capability for heterogeneous interactions. HGSolver can handle tasks with inconsistent input and output geometries, such as super-resolution and domain reconstruction problems. The model supports forward, inverse, and reconstruction tasks.

**Strengths:**

The paper proposes a unified projection perspective on physical attention, redefining physical states as projections onto learnable function bases in Hilbert space. This provides a solid mathematical foundation for understanding physical attention mechanisms, elevating traditional attention mechanisms to the level of function space analysis.

**Weaknesses:**

1. State-of-the-art performance results on the  inverse, reconstruction, and forward tasks are claimed in this paper. However, Table 1 only presents an ablation study on the physics attention mechanism. The authors did not conduct comparative experiments with other methods that support decoupled inputs and outputs, such as LNO.
2. In the experiments in Table 3, only the variant TransolverXP is included, while the results of HGSolver are absent.

**Questions:**

Since HGSolver supports Heterogeneous Geometry tasks with decoupled inputs and outputs, it should also support Homogeneous Geometry tasks. Why is a specialized TransolverXP variant designed specifically for Homogeneous Geometry tasks? What are the specific structural differences between TransolverXP and HGSolver in terms of design?

---

### Note · Authors · 2025-11-25

I have read and agree with the venue's withdrawal policy on behalf of myself and my co-authors.